# Exploring the use of pulsed erbium lasers to retrieve a zirconia crown from a zirconia implant abutment

**Ahmed Elkharashi[1], Kinga Grzech-Leśniak[2], Janina Golob Deeb[3], Aous A. Abdulmajeed[4], Sompop Bencharit** [4,5,6] *

**1** Department of Biochemistry and Molecular Biology, School of Medicine, Virginia Commonwealth University, Richmond, Virginia, United States of America, **2** Department of Oral Surgery, Wroclaw Medical University, Wroclaw, Poland, **3** Department of Periodontology, School of Dentistry, Virginia Commonwealth University, Richmond, Virginia, United States of America, **4** Department of General Practice, School of Dentistry, Virginia Commonwealth University, Richmond, Virginia, United States of America, **5** Department of Oral and Maxillofacial Surgery, School of Dentistry, Virginia Commonwealth University, Richmond, Virginia, United States of America, **6** Department of Biomedical Engineering, College of Engineering, Virginia Commonwealth University, Richmond, Virginia, United States of America

* sbencharit@vcu.edu

**Data Availability Statement:** All relevant data are within the paper.

**Funding:** Fotona had loaned us the Er:YAG laser to be used in the study. However, the company did

## Abstract

### Background

Removal of cement-retained implant fixed restorations when needed, can be challenging. Conventional methods of crown removal are time consuming and costly for patients and practitioners. This research explored the use of two different types of pulsed erbium lasers as a non-invasive tool to retrieve cemented zirconia crowns from zirconia implant abutments.

### Materials and methods

Twenty identical zirconia crowns were cemented onto 20 identical zirconia prefabricated abutments using self-adhesive resin cement. The specimens were divided into two groups for laser assisted crown removal; G1 for erbium-doped yttrium aluminum garnet laser (Er: YAG), and G2 for erbium, chromium-doped yttrium, scandium, gallium and garnet (Er,Cr: YSGG). For the G1, after the first crown removal, the specimens were re-cemented and removed again using the Er:YAG laser. Times needed to remove the crowns were recorded and analyzed using ANOVA ($\alpha = 0.05$). The surfaces of the crown and the abutment were further examined using scanning electron microscopy (SEM) and energy-dispersive X-ray spectroscopy (EDS) analyses.

### Results

The average times of zirconia crown removal from zirconia abutments were 5 min 20 sec and 5 min 15 sec for the Er:YAG laser of first and second experiments (G1), and 5 min 55 sec for the Er,Cr:YSGG laser experiment (G2). No statistical differences were observed

not support us financially and had no role in study design, data collection and analysis, decision to publish, or preparation of the manuscript.

**Competing interests:** Fotona had loaned us the Er: YAG laser to be used in the study. However, the company did not support us financially and had no role in study design, data collection and analysis, decision to publish, or preparation of the manuscript. This does not alter our adherence to PLOS ONE policies on sharing data and materials.

among the groups. SEM and EDS examinations of the materials showed no visual surface damaging or material alteration from the two pulsed erbium lasers.

## Conclusions

Both types of pulsed erbium lasers can be viable alternatives for retrieving a zirconia crown from a zirconia implant abutment. Despite operating at different wavelengths, the Er:YAG and Er,Cr:YSGG lasers, perform similarly in removing a zirconia crown from a zirconia implant abutment with similar parameters. There are no visual and elemental composition damages as a result of irradiation with pulsed erbium lasers.

## Introduction

Zirconia restorative materials have become one of the most commonly used materials in restorative dentistry [1–4] and are often a treatment of choice for a single-tooth implant in the esthetic zone. [5–8] Despite advantages of zirconia, the removal of the cemented zirconia crowns off their fitting abutments when needed can be difficult. [2,9] Zirconia is considered to be the strongest ceramic dental material currently available. Cement-retained zirconia implant crowns are therefore difficult to remove using conventional methods such as rotary handpieces with cutting diamond burs. This conventional method not only takes a considerable amount of time, but also leaves the zirconia crown un-reusable and often damage the zirconia abutment. [10,11]

Previous studies have proven the possible use of pulsed erbium family lasers such as erbium-doped yttrium aluminum garnet laser (Er:YAG) and erbium, chromium-doped yttrium, scandium, gallium and garnet (Er,Cr:YSGG) lasers to retrieve restorations made of different ceramic and composite resin materials from natural teeth [12–17] and various types of implant abutments. [10,11,18,19] Recent studies have proven the validity and safety of using Er:YAG laser to remove lithium disilicate crowns from titanium and zirconia abutments. [10,11] However, the use of pulsed erbium lasers to remove cemented zirconia crowns off zirconia abutments is yet to be explored.

Pulsed erbium lasers are generally high-power lasers that can perform a wide range of functions, such as skin resurfacing and soft tissue de-epithelialization, [20] ablative and non-ablative, [21] vaporizing soft tissue as means of bacterial control, [22–24] and in cosmetic surgery to removal of skin epidermis to accelerated healing and reduce scars [22,25]. This study proposed that erbium pulsed lasers such as Er:YAG with 2940 nm wavelength and Er,Cr:YSGG with 2780 nm wavelength can be used to remove a zirconia crown off a zirconia implant abutment. Theoretically, the wavelengths of these lasers operate with the mid-infrared spectrum, which coincides with the range for water absorption spectrum. Hence, water molecules and possibly the remaining monomer molecules in the cement are the target chromophore for these erbium pulsed lasers. [11–14,16] Water and remaining monomer molecules trapped within the luting cement polymerized structure are the main target chromophore for the purpose of removing a crown. These molecules absorb the wavelength emitted from the laser, and release an energy that is destructive to cement polymerized structure. [10,11]

This study utilized two types of pulsed erbium lasers, Er:YAG and Er,Cr:YSGG lasers to retrieve zirconia crowns from zirconia abutments. The study aimed to compare the two laser efficacies, in terms of crown removing times, under similar parameters. The performance of Er:YAG laser was also tested over a repeated cementation trial simulating clinical scenarios of

reusing the crown via repetitive cementation. Inspections for surface damage following irradiations were performed through scanning electron microscopy (SEM) analysis. Material composition as a response to irradiation was also examined using energy-dispersive X-ray spectroscopy (EDS).

## Materials and methods

Specimen design and fabrication process was similar to previous studies [10,11]. A dental cast of a patient missing a mandibular left second premolar that was replaced with a single dental implant (4.5 mm platform Tapered Screw-Vent Implant, Zimmer Biomet) was used. A zirconia prefabricated implant abutment (Contour Abutment, Zimmer Biomet) was placed onto the study cast. The abutment and the cast, the opposing cast and the buccal interocclusal registration were scanned using an intraoral scanner (Emerald, Planmeca). These scans were then used to design a crown using Romexis version 5 software (Planmeca). The crown was designed to have approximately 1–2 mm thickness except in the crown margin, which was about 0.5 mm thick. The design was then exported in the standard tessellation language (STL) format and zirconia crowns were fabricated using a milling machine (PrograMill PM7, Ivoclar Vivadent). Twenty identical zirconia monolithic crowns were milled using a monolithic disc (IPS e.max ZirCAD, MT Multi, Ivoclar Vivadent). The zirconia used in this study is a combination of 3 mol% yttria stabilized tetragonal zirconia polycrystal (3Y-TZP) and 5 mol% yttria stabilized tetragonal zirconia polycrystal (5Y-TZP).

The milled crowns were then sintered for 9 hours and 50 minutes per the manufacturer's recommendation using a sintering furnace (Programat S1 1600, Ivoclar Vivadent). The sintered crowns were air-abraded using 50 μm aluminum oxide particles. Then the crowns underwent characterization and staining process using the same furnace (Programat S1 1600, Ivoclar Vivadent). The glazed crowns were then again air-abraded with 50 μm aluminum oxide particles, and the excess glaze if any around the crown margin was removed and polished. The crowns were then tried on the abutment to determine the fitting.

The abutments were then installed onto the implant fixtures. A piece of polytetrafluoroethylene (PTFE, Teflon) tape was used to cover the screw access of the abutment. Prior to cementation, the primer (Monobond Plus, Ivoclar Vivadent) was applied to the intaglio surface of the crown and left on for 60 seconds before blow drying. The crowns were luted onto the abutments using self-adhesive resin cement (RelyX™ Unicem 2, 3M). Buccal, lingual, and occlusal surfaces of the crown were subjected to a 1-to-2 second period of light polymerization to facilitate removal of excess cement. The cemented crowns were left under finger pressure for 6 minutes for complete polymerization. The cemented crowns were kept for 24 hours in a 100% humidifying chamber prior to the crown removal experiment. A total of 20 crowns cemented on abutment-implant fixture specimens were made.

Two pulsed erbium lasers were used in this study, erbium-doped yttrium aluminum garnet laser (Er:YAG) with 2940 nm wavelength (LightWalker, Fotona), and erbium, chromium-doped yttrium (LightWalker AT / AT S, Fotona), scandium, gallium and garnet (Er,Cr:YSGG) with 2780 nm wavelength (Waterlase iPlus, Biolase). Er:YAG laser was set at: 300 mJ, 15Hz, 4.5 W, operation mode: SSP mode (50 μs pulse duration) 2 water/2 air, while Er,Cr:YSGG laser was set at: 4.5 W, 15 Hz, with operation mode: H (60 μs pulse duration), 20 water/20 air. The irradiation protocol was the same for both lasers, and similar to the protocol from the previous study [10,11]. Attempts were made to standardize the laser irradiation and crown removal process. For each experiment, same three investigators performed each of the following procedures for each crown removal. One investigator did all laser irradiation. One investigator attempted to remove the crown. And one investigator recorded the removal time."

The crown specimens were grouped relative to the type of laser used for the crown removal and the number of times they were recemented. Group 1 (G1) was comprised of 10 crowns that had undergone two successive crown removal trials with recementation via the Er:YAG laser, with subgroups Er1 (Er:YAG cementation & 1st irradiation) and Er2 (Er:YAG recementation & 2nd irradiation). Group 2 (G2) consisted of a different set of 10 new crowns that were cemented and irradiated once using the second laser, Er,Cr:YSGG laser. Note that an extra zirconia crown was made. This extra zirconia crown and an extra prefabricated zirconia abutment were used in the SEM/EDS analysis as control samples.

The cemented crown/abutment/implant was placed in a stereolithographic printed typodont model with adjacent natural teeth mounted to simulate a clinical situation.(Fig 1) The lasers were oriented perpendicular to the crown surface using the non-contact method (~5–10 mm away from crown surface). The cooling air/water spray feature was used during the entire irradiation time. The irradiation was carried out through directing the laser axially onto all non-occlusal surfaces for 180 seconds while rotating the crown slowly, then 60 seconds onto the occlusal surface, then lastly 30 seconds irradiation of all crown surfaces. After the initial 240 seconds of irradiation, the crown's dislodgement was assessed through gentle tapping and pulling action. If the crown was not dislodged, subsequent extra 30 seconds of additional irradiation was administered and an additional attempt at crown's dislodgement was made. This latter process was repeated until the crown was retrieved. During the irradiation, temperatures

**Fig 1. Study workflow demonstrating the laser crown removal experiment and the SEM/EDS analysis of the specimen.** The laser was applied to the cameo surface of the crown prior to attempt to remove the crown using finger pressure. The abutment and crown specimens were then subjected to SEM/EDS analysis. Different areas of the abutment specimen with less and more cement remaining were subjected to EDS elemental composition analysis.

of the crown, abutment, and implant fixture were also monitored similar to previous studies. [10,11,18]

After the first experiment for the Er:YAG group (G1 Er1), the crown specimens (n = 10) were reused (G1 Er2) after their first successful crown removal to mimic the repeated clinical cementation. The leftover cement in these crowns' fitting surface was cleaned off via air-abrasion using 50 μm aluminum oxide particles and at a pressure of 2 pound per square inch (psi). The crowns were then re-cemented onto the zirconia abutments following the same initial cementing protocol and kept for 24 hours in 100% humidifying chamber prior to their consecutive crown removal attempt.

The total crown removal time for each crown in both groups was recorded and calculated by adding each extra 30 seconds needed to retrieve the crown to the initial 240 seconds of irradiation. The crown removal times were then analyzed for statistical significance using one-way analysis of variance (ANOVA: single factor, α = 0.05).

To test for structural integrity and possible surface damage to the crown and abutment due to irradiation, the specimens were analyzed by SEM and EDS technologies (Fig 1). After the crown removal experiment, the underlying intaglio surface as well as the cameo surface of the crown and the cameo surface of the abutment were inspected using scanning electron microscopy (SEM) analysis (JEOL 6610LV, JEOL, Japan). Energy-dispersive X-ray spectroscopy (EDS) analysis was also used to examine the variations in the elemental composition of the crown, abutment as well as remaining cement if any. An extra zirconia crown and an extra zirconia abutment that had not gone through cementation and irradiation processes were used as control specimens. The specimens were not coated and EDS was performed using a low vacuum mode in SEM with 20 kV energy range.

## Results

The average times needed to remove a zirconia crown from a zirconia implant abutment were 5 min 20 sec for G1 Er1 (first cementation and Er:YAG retrieval), 5 min 15 sec in G1 Er2 (second cementation and Er:YAG retrieval), and 5 min 55 sec in G2 (cementation and Er,Cr: YSGG retrieval). Overall, across all groups and subgroups there were no statistical differences observed in the removal times according to the ANOVA statistical analysis (p = 0.32). (Table 1, S1 Table) When irradiation applied to the cervical part of the crown from 1 to 10 minutes, the temperature ranges of the crown, abutment and implant fixtures were 21.3˚C to 27.7˚C, 19.9˚C to 26.4˚C, and 22.1˚C to 27.4˚C, respectively. Note that the temperatures of the water ranged from 18.6˚C to 21.4˚C.

Examination using SEM showed no major structural changes or damage suggestive of photoablation or thermal ablation of the abutments (Fig 1). The remaining cement appeared to stay on the abutment surface more than the crown surface. Additionally, no carbonization

**Table 1. The average (mean) and standard deviations (SD) of crown removal time.**

| Grouping[$] | G1 [Er1] | G1 [Er2] | G2 |
|---|---|---|---|
| Mean | 312 | 309 | 333 |
| SD | 42.90 | 42.54 | 26.27 |
| **ANOVA**[*] | P<0.05 | | |
| **P Value** | 0.32 | | |

[$]G1 [Er1]: Er:YAG cementation & 1st irradiation, G1 [Er2]: Er:YAG re-cementation & 2nd irradiation, G2: Er,Cr: YSGG cementation & irradiation.

[*]Level of statistical significance, p<0.05.

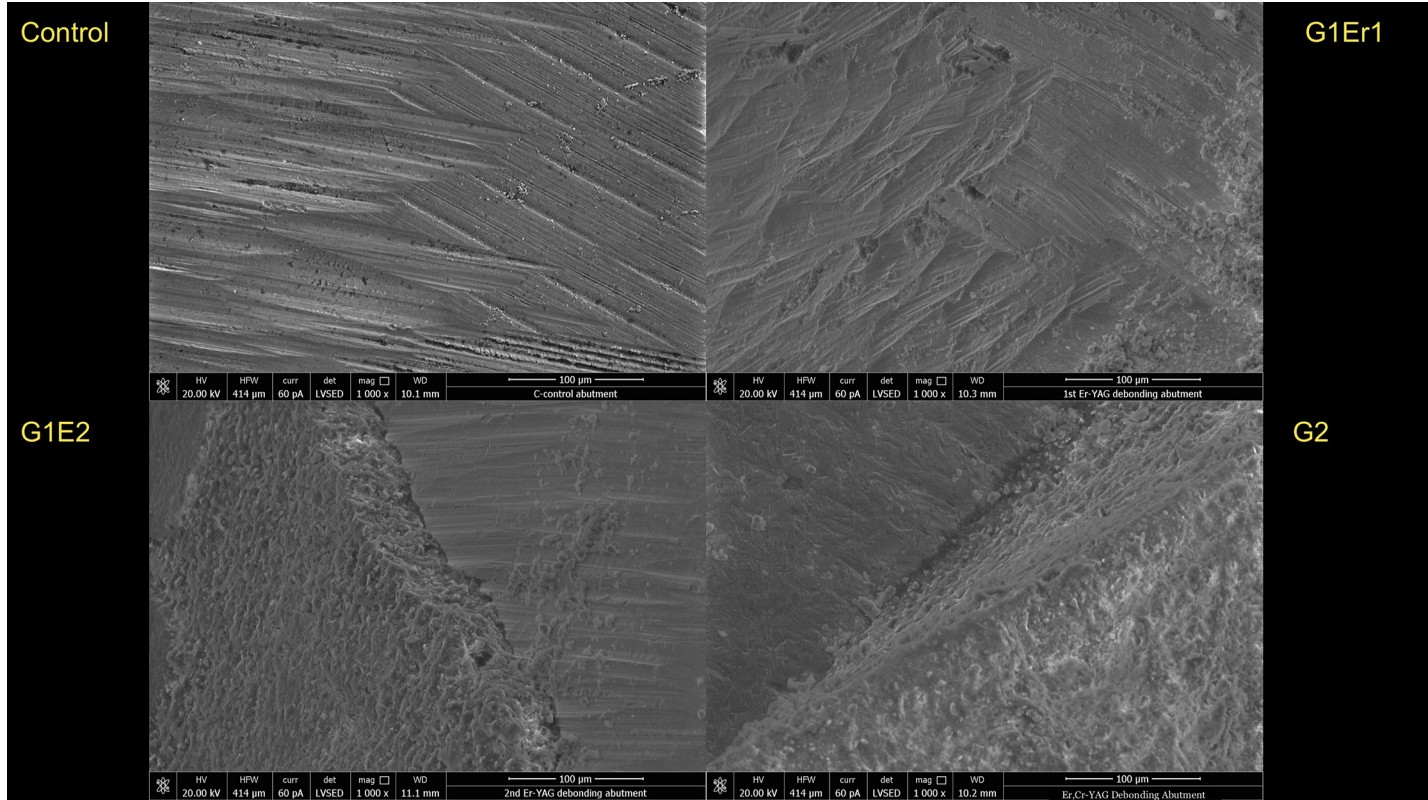

**Fig 2. SEM analyses of zirconia abutments.** Control abutment with no laser exposure demonstrating the milled zirconia surface without any cement (Top Left), Er: YAG lasered zirconia abutment after first cementation (G1 Er1) with some remnant cement demonstrating similar zirconia surface (Top Right), Er:YAG lasered zirconia abutment after repeated cementation (G1 Er2) demonstrating zirconia surface and remnant cement (Bottom Left), and Er,Cr:YSGG lasered zirconia abutment (G2) with some remnant cement (Bottom Right). Note that more cement remained in the G1 Er2 and G2 groups.

on the zirconia implant abutment or crown was observed. SEM analysis conducted to test the damage of the implant crown and abutment surface showed that all laser-assisted crown removal samples demonstrated no major noticeable cracks or fractures with macro and microstructure. Slight partial ablation of the cement during irradiation was occasionally observed. All the abutment groups demonstrated similar surface roughness and characters. The control abutment, that had no laser exposure or cementation, showed a clearly smoother surface than other samples with no cement remaining. The G1 Er1 group showed little cement remaining. The G1 Er2 and G2 groups showed more cement remaining than the G1 Er1 (Fig 2). For the crowns, the cameo surface of the control group appeared to be the smoothest. The cameo surfaces of the test samples, G1 Er1, G1 Er2 and G2, appeared to be slightly rougher suggesting the roughness increased from the irradiated glazed feldspathic surface. The intaglio surfaces of all groups appeared to be similar in the roughness. However, the first Er:YAG group showed less cement compared to the repeated Er:YAG and Er,Cr:YSGG groups. (Fig 2) In the EDS mode, the intaglio surfaces of the crown and the cameo surface of the abutment with and without remaining cement were analyzed. (Figs 3, 4 and 5) The EDS analyses were performed in three areas, the abutments with more and less cement remaining, and the crown. The EDS spectra and elemental compositions were very similar among the three groups in the less cement abutments and the crowns. However, the EDS analysis of the abutment with more cement remaining areas demonstrated a different spectra and elemental compositions for Er, Cr:YSGG compared to the Er:YAG groups perhaps from more cement remaining.

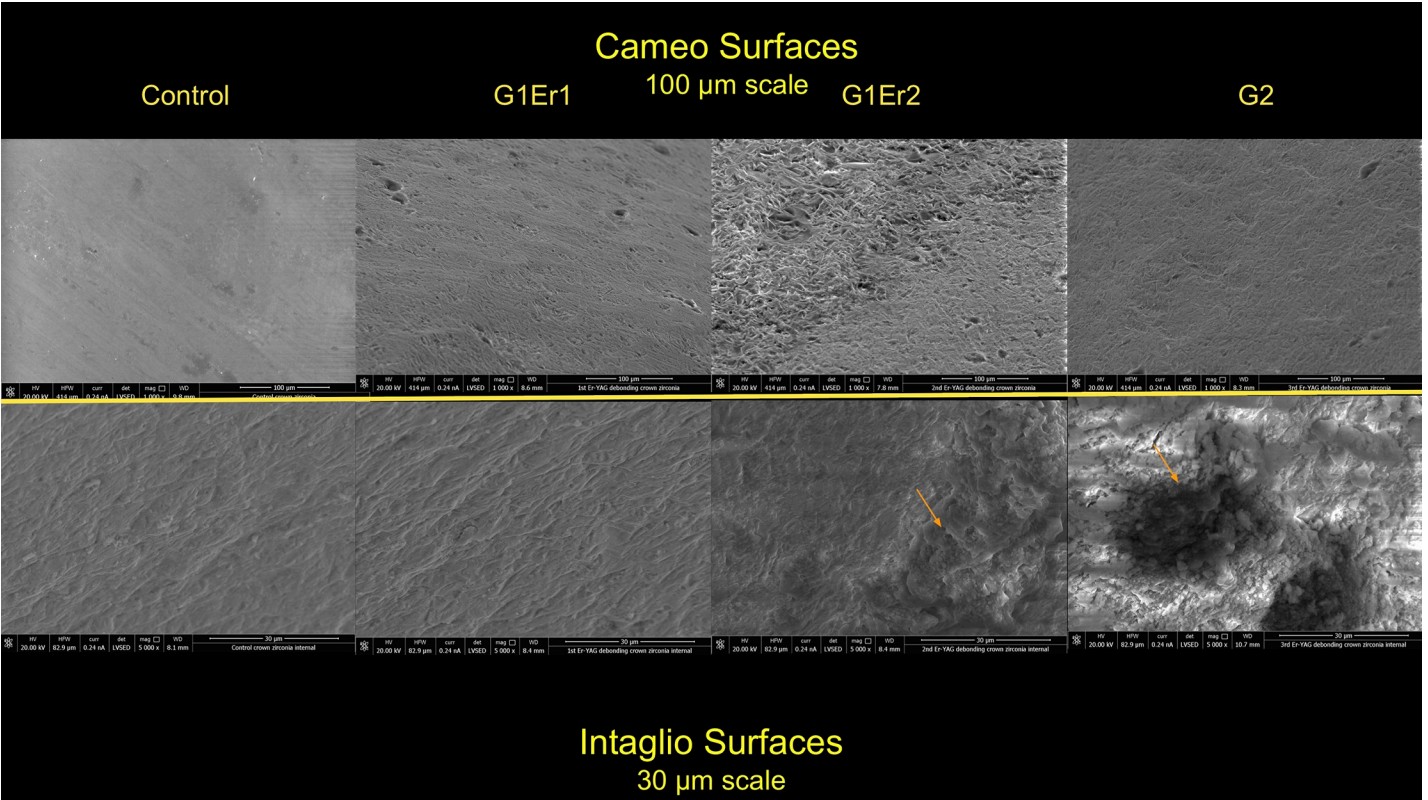

**Fig 3. SEM analyses of zirconia crown.** (Top from left to right) SEM images of control, G1Er1, G1Er2, and G2 on the cameo surface. Notice surface roughness of cameo surface from smoothest to roughest: Control < G1Er1/G2 < G1E2. (Bottom from left to right) SEM images of control, G1Er1, G1Er2, and G2 on the intaglio surface. Cement accumulation from no cement to most cement: Control < G1Er1 < G1Er2 < G2. Arrows show the area of cement remaining in the intaglio surface of G1Er2 and G2.

## Discussion

The results suggest that erbium pulsed lasers, Er:YAG and Er,Cr:YSGG, can not only facilitate the removal of zirconia crowns from zirconia implant abutments within a short period of time (~ 5 to 6 minutes on average), but also offer a more conservative and less invasive treatment approach. Therefore, erbium pulsed lasers should be considered as a valuable option especially when retrieval of cement-retained implant restorations are indicated. Laser-assisted crown removal times were recorded to be around 5 minutes for both lasers with the temperature ranges of ~21˚C to 28˚C was about 2˚C higher than previous report of removing a lithium disilicate crown from a titanium or zirconia implant abutment. [10,11] This may due to the different crown materials. There were no statistical differences when comparing the crown removal times of the two lasers, nor when comparing the crown removal times for the same laser, Er:YAG, after repeated cementation. This suggests that the two erbium pulsed lasers may have similar mechanisms and may be used interchangeably to retrieve a zirconia implant crown. The two lasers having similar crown removal times despite having different wavelengths can be attributed to the fact that both lasers' wavelength fall in the mid-infrared spectrum where water remains the target chromophore.

Er:YAG and Er,Cr:YSGG lasers while both are erbium pulsed lasers, they have different pulse mechanisms in energizing the flashlamp, source of the laser energy. Er,Cr:YSGG uses the older technology known as Pulse Forming Network (PFN), while Er:YAG uses the newer technology known as Variable Square Pulse (VSP). [26] PFN pulses have a bell shape with usually

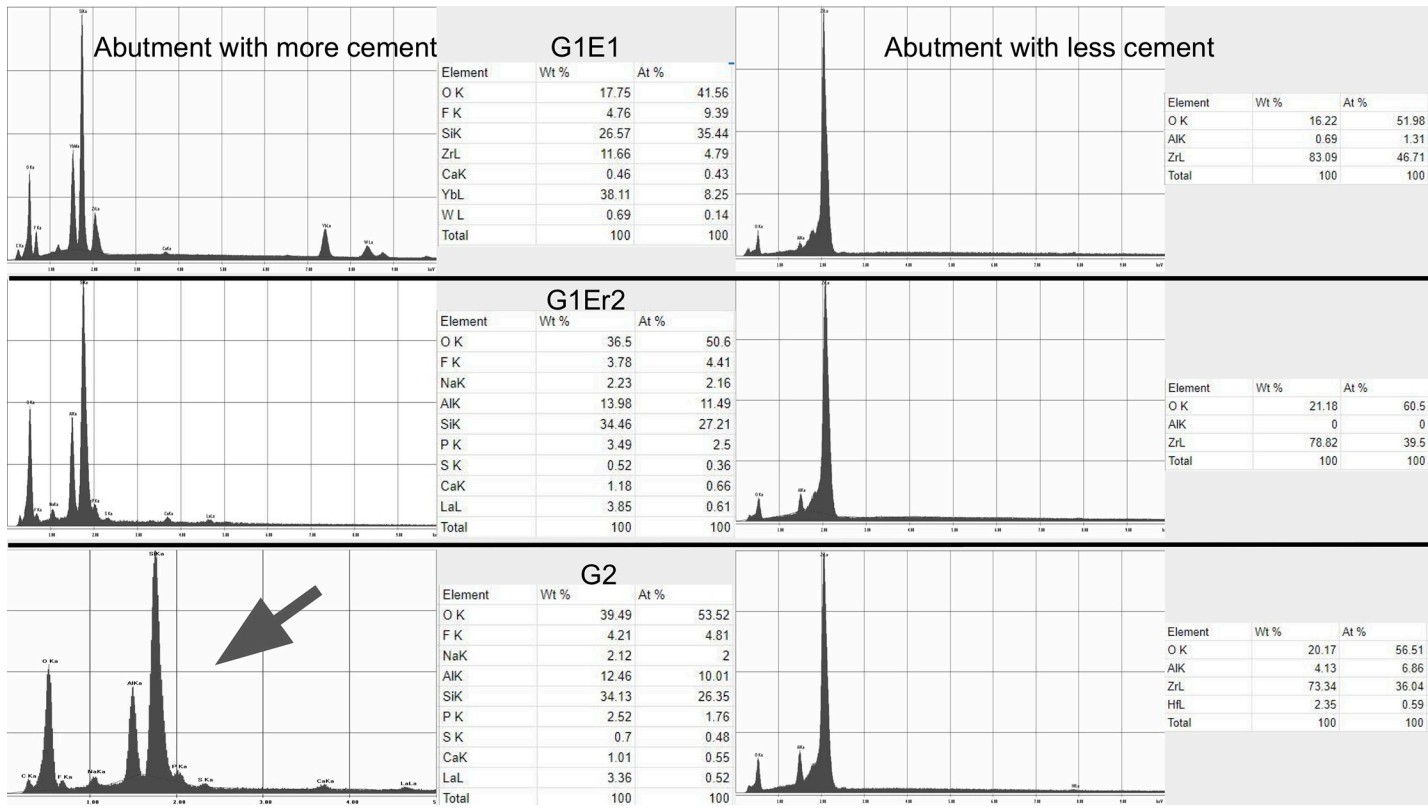

**Fig 4. EDS analyses of zirconia abutments.** Spectra and elemental composition of abutment with first cement Er:YAG laser (G1 Er1) for more cement (Top Left) and less cement area (Top Right); spectra and elemental composition of abutment with repeated cement Er:YAG laser (G1 Er2) for more cement (Middle Left) and less cement area (Middle Right); and spectra and elemental composition of abutment with cement Er,Cr:YSGG laser (G2) for more cement (Bottom Left) and less cement area (Bottom Right). While all less cement abutment compositions were similar, there seemed to be more cement in G2 group as per different spectra/elemental compositions compared to other groups (Arrow).

fixed duration. VSP pulses have a square shape with variable pulse duration. While the energy pulses may seem to be similar because of the different pulsed energy generated, it may behave differently in the oral hard tissues or different dental materials. [27] In general, it appears that Er,Cr:YSGG laser is less effective but penetrates deeper and not as localized as Er:YAG in the dentin. [27–30]

A previous study that utilizes the same Er:YAG laser showed that the time required to remove lithium disilicate crowns from zirconia abutments was considerably shorter, relative to the zirconia crowns used in this study. [10,11] The required time to retrieve lithium disilicate crowns from zirconia abutments was approximately 3 minutes, [10,11] compared to approximately 5 minutes for zirconia crowns in this study. The more crystalline structure of zirconia compared to lithium disilicate may interfere with laser penetration. This ~5 min removal time of a zirconia implant crown is also similar to a previous study using Er:YAG laser to remove a zirconia crown from a natural tooth abutment. [12] Therefore, it is suggested that dentists should factor in the type of material of a restoration used when estimating the time needed to dislodge fixed dental prosthesis if frequent removals of the particular prostheses is expected. Interestingly, when compared to the previous study, this study showed consistent results for crown removal times after short-term repeated cementation. [11] New cementation and repeated cementation of zirconia crowns to zirconia abutments experienced minimal change for the required time for crown removal using Er:YAG laser. Note that the reason why an

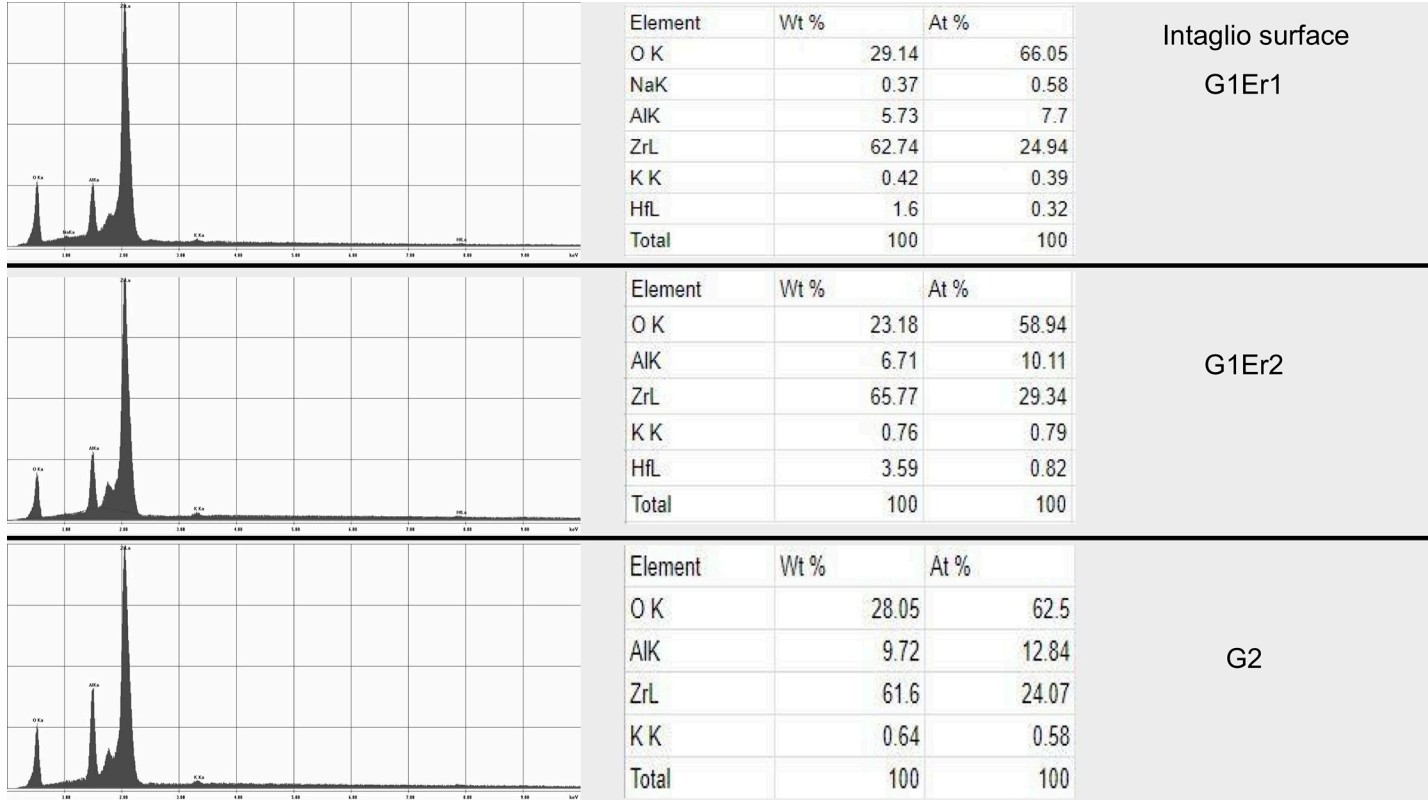

**Fig 5. EDS analyses of zirconia crowns.** Spectra and elemental composition of abutment with first cement Er:YAG (G1 Er1) (Top), repeated cement Er:YAG (G1 Er2) (Middle), and Er,Cr:YSGG (G2) (Bottom) groups. All demonstrated similar spectra and elemental compositions.

equal experiment for both Erbium lasers was not planned was there were more previous experimental data and research on the Er:YAG laser in implant crown removal. However, at the time of designing this experiment there was no information in the literature on the Er,Cr:YSGG laser implant crown removal. The investigators were not certain if Er,Cr:YSGG would work in the similar way as Er:YAG. Certainly, there will be in the future, more experiments with different materials and repeated cementations on Er,Cr:YSGG laser implant crown removal.

This study also showed that pulsed erbium lasers do not damage the retrieved prosthesis in the process, as the option to reuse retrieved restoration is a major advantage over conventional methods. SEM/EDS analyses for this study following Er:YAG and Er,Cr:YSGG irradiation showed no visual damage to any crown or abutment nor any macro- or micro- structural surface damage consistent with previous Er:YAG studies for lithium disilicate crowns/zirconia- or titanium implant abutments. [10,11] SEM/EDS analyses have been used to examine zirconia/porcelain veneer interface [31] as well as zirconia bonding to self-adhesive resin cement. [32] EDS analyses provide additional insights into the elemental composition of irradiated zirconia abutments and crowns that seems to have no damaging effect from erbium laser irradiation. However, slight increased surface roughness of the cameo surface of the crown was seen as a result of laser exposure. It is possible that that the laser may have roughen the feldspathic porcelain similar to other studies. [33]

A study using Er,Cr:YSGG laser to remove lithium disilicate crowns from natural teeth, showed a lesser time of removal (~1–3 min) [16,39] This may have been a result of a different

type or thickness of restorations, different type of cements, and a higher energy setting. Previous studies suggested that the Er,Cr:YSGG laser is less efficient compared to Er:YAG. [16,34,35] In this current study, the zirconia crown removal times were slightly longer for the Er,Cr:YSGG group compared to the Er:YAG without statistical significance. The Fourier Transform Infrared Spectroscopy (FTIR), method used to measure material energy absorption, demonstrates a broad $H_2O$/OH absorption band of wavelengths in the range of 3,750–3,640 and 3,600–3,400 nm. [34,35] Similarly, composite resin cements, such as Multilink (Ivoclar Vivadent) demonstrate a distinct absorption peak at ~3,401 nm. These ranges coincide with the erbium pulsed laser emission wavelength where there was little radiation absorption to zirconia material. [12–14] Thus, this allows debonding via irradiation energization of cement with little or no damage to the crown materials.

While this study showed minor physical or composition changes of the zirconia material, the effects from laser induced changes of the material surface at small micro- and nano-scale remain unclear. There was more cement remaining in the crown and abutment of the repeated cement Er:YAG and Er,Cr:YSGG groups in the current study. The SEM observation was also confirmed by the EDS analyses that Er,Cr:YSGG group demonstrated most cement remaining on the crown intaglio surface. Some evidence suggested that the zirconia treatments with Er: YAG and Er,Cr:YSGG improve surface roughness and likely increase the bond strength to the resin cement. [36–39] The improved roughness and bond strength to the resin as a result of laser treatment at the pre-sinter stage are significant and maybe an alternative to air abrasion. The laser effects after the sintering process appear to have only limited or no improvement of resin bonding strength. [36,40,41] The Er:YAG and Nd:YAG laser treatments can, at higher energy settings than this study, produce microcracks in the zirconia that may improve retention with the resin cement. [36]

There are some limitations to the study. First, limitations of an *in vitro* study are recognized. While we added typodont and applied laser in the phantom head mimicking the patient and improved the protocol from previous studies, [10,11,18] it is likely that applications of these two lasers intraorally may not be as effective due to access limitations such as cheek, tongue, limited mouth opening. Second, it is not known how the laser may affect soft tissue attachment or microbial biofilm adherence. While the microroughness of the sintered monolithic zirconia is minimal, the nanoroughness of zirconia material as well as the roughness of the feldspathic porcelain glaze may alter soft tissue attachment or biofilm adherence. Majority of literature on removing biofilm with Er:YAG laser suggest that it is effective in removing biofilm with no change in the zirconia dental implant structures. [42–45] However, Er,Cr:YSGG laser may have some effects on the roughness of zirconia surface. [44] Further analyses of both lasers in the biological system or *in vivo* are therefore needed. Third, there is no repeated cementation for Er,Cr:YSGG laser group in this current study. Since the Er:YAG has been studied more thoroughly on the removal of implant crowns from implant abutments, in the current study Er,Cr:YSGG was being explored and compared. Future studies including the temperature changes with different types of crown/abutment materials and cements will be needed for Er, Cr:YSGG laser.

## Conclusions

Repeated cementation of a zirconia crown onto a zirconia abutment following irradiation using Er:YAG laser, had no effect on the crown removal time. When comparing the crown removal times for the two pulsed erbium lasers, Er:YAG and Er,Cr:YSGG, no statistically significant difference was observed. Lastly, upon examination with SEM and EDS no surface structure damage was observed nor a change in material composition was experienced

following Er:YAG irradiation. Both Er:YAG and Er,Cr:YSGG lasers are effective and non-invasive tools for retrieving a zirconia crown from an zirconia implant abutment.

## Supporting information

**S1 Table. Experimental crown removal time and statistical analysis.**
(XLSX)

## Acknowledgments

We would like to thank our exchange dental students from Slovenia, Domen Kanduti and Lenart Škrjanc, who had helped us with the research methods. Special thanks to the faculty and staff of Virginia Commonwealth University Center of Digital Dentistry, especially Marithe Blacagon for specimen preparation. Thanks also to Dr. Janina Lewis and Dr. Robert F. Diegelmann, for providing inputs for the research.

## Author Contributions

**Conceptualization:** Kinga Grzech-Leśniak, Janina Golob Deeb, Sompop Bencharit.

**Data curation:** Ahmed Elkharashi, Kinga Grzech-Leśniak, Sompop Bencharit.

**Formal analysis:** Ahmed Elkharashi, Sompop Bencharit.

**Funding acquisition:** Sompop Bencharit.

**Investigation:** Ahmed Elkharashi, Kinga Grzech-Leśniak, Janina Golob Deeb, Aous A. Abdulmajeed, Sompop Bencharit.

**Methodology:** Ahmed Elkharashi, Kinga Grzech-Leśniak, Aous A. Abdulmajeed, Sompop Bencharit.

**Project administration:** Ahmed Elkharashi.

**Resources:** Sompop Bencharit.

**Supervision:** Sompop Bencharit.

**Visualization:** Sompop Bencharit.

**Writing – original draft:** Ahmed Elkharashi, Sompop Bencharit.

**Writing – review & editing:** Ahmed Elkharashi, Kinga Grzech-Leśniak, Janina Golob Deeb, Aous A. Abdulmajeed, Sompop Bencharit.

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
