## [Decision Letter · Decision Letter 0]

15 Apr 2020

PONE-D-20-08097

Exploring the use of pulsed erbium lasers to retrieve a zirconia crown from a zirconia implant abutment

PLOS ONE

Dear Dr. Bencharit,

Thank you for submitting your manuscript to PLOS ONE. After careful consideration, we feel that it has merit but does not fully meet PLOS ONE’s publication criteria as it currently stands. Therefore, we invite you to submit a revised version of the manuscript that addresses the points raised during the review process.

We would appreciate receiving your revised manuscript by May 30 2020 11:59PM. To enhance the reproducibility of your results, we recommend that if applicable you deposit your laboratory protocols in protocols.io, where a protocol can be assigned its own identifier (DOI) such that it can be cited independently in the future. For instructions see: http://journals.plos.org/plosone/s/submission-guidelines#loc-laboratory-protocols

We look forward to receiving your revised manuscript.

Kind regards,

Rafael Sarkis-Onofre

Academic Editor

PLOS ONE

Journal Requirements:

2.Thank you for stating the following financial disclosure:"N/A"

Please provide an amended Funding Statement that declares *and fully names all* the funding or sources of support received during this specific study (whether external or internal to your organization) as detailed online in our guide for authors at http://journals.plos.org/plosone/s/submit-now.  

Please state what role the funders took in the study.  If any authors received a salary from any of your funders, please state which authors and which funder. If the funders had no role, please state: "The funders had no role in study design, data collection and analysis, decision to publish, or preparation of the manuscript."

Reviewers' comments:

Reviewer's Responses to Questions

**Comments to the Author**

1. Is the manuscript technically sound, and do the data support the conclusions?

Reviewer #1: Yes

Reviewer #2: Yes

2. Has the statistical analysis been performed appropriately and rigorously? 

Reviewer #1: Yes

Reviewer #2: Yes

3. Have the authors made all data underlying the findings in their manuscript fully available?

Reviewer #1: Yes

Reviewer #2: No

4. Is the manuscript presented in an intelligible fashion and written in standard English?

Reviewer #1: Yes

Reviewer #2: Yes

5. Review Comments to the Author

Reviewer #1: The main topic is interesting and could result in the retrievability of implant-retained cemented crowns, which is very interesting considering clinical practice. Although the journal guidelines do not have restrictions on word count, number of figures, or amount of supporting information, it states that authors are encouraged to present and discuss their findings concisely. Thus, my main consideration regarding this manuscript is that it should be thoroughly revised in order to provide only essential information to the readership. The way it is presented is too long and sometimes repetitive; some parts of the introduction should be moved to the discussion section and also some parts of the discussion section should be removed or revised in order to be more concise.

Regarding to the presentation of the results, Table 1 does not provide a title and it does not present which outcome was measured and compared in it.

Figures 1A and 1B does not add much to the text. I suggest the authors create a scheme or use another picture to illustrate the use of the laser to remove the crowns. Also, the high number of figures should be revised by the authors. I think that there is no need for all those SEM and EDS images since no significant difference was observed when groups were compared.

Reviewer #2: Thank you for considering PLOS ONE to publish your work. Authors explored the use of two different types of

pulsed erbium lasers as a non-invasive tool to retrieve cemented zirconia crowns from zirconia implant abutments. At one of the lasers authors explored additionally its use at a second removal of the crown that was recemented. Basically the manuscript deals with an interesting topic, clinically relevant, the methods seems sound, the results interesting, and authors did a great job discussing and introducing the theme. By that, in my humble opinion, the manuscript has potential to be published as it contributes to the knowledge of the theme. Despite that some aspects should be considered further, prior to acceptance:

Major aspects:

- PLOS ONE requires that authors to enable access to the data, authors answered that the data is within the manuscript, however only mean and standard deviation could be found on table1. Please make the data available, or attempt on justify its absence.

- Authors should at least justify why exploring two scenarios of crown retrieval using one laser only, and not with both methods under study.

- One important aspect is in regards of sample size estimation. It can be seen that using G1 and G2 the Standard deviation was higher than at G3. Perhaps, with a large sample size it would be enable to see statistical diferences. It is understandable that clinically a time ranging from 5-6min did not correspond to great/ clinically relevant differences. Despite that, it is necessary to justify the sample size by means of an estimation or a power analysis, and discuss such aspect.

- The researcher that attempt on remove the crown was blind for the group? Authors describes as pulling action and gentle tapping, which is rather unclear. There was any standardisation of the load applied during this procedure? Or any assurance of been applied at the same manner for all groups.

- Authors expressed that the Y-TZP used is partially 3Y-TZP, and partially 5Y-TZP. 3Y-TZP is known as a partially stabilised zirconia, and by that phase transformation may happen, and potentially low-temperature degradation through time. Thus, one important aspect in regards of potential structural alterations promoted by laser application would be it triggering phase transformation. Even that authors did not access such outcome by raman or X-RD analysis, this aspect should be at least discussed as a limitation. OR the partially composition of 5Y-TZP makes this material fully stable and the phase transformation did not takes place?

Minor aspects:

- At the abstract the groups description is very confusing. At the manuscript material and methods section is much more clear. Please review.

- Figures, if possible to add group codenames on figures i believe it would facilitate for the reader. Now it is only present on Figure legend description.

- Figures, the amount of figures seems to be excessive. All of these figures are really necessary?

6. PLOS authors have the option to publish the peer review history of their article (what does this mean?). If published, this will include your full peer review and any attached files.

Reviewer #1: No

Reviewer #2: No

---

## [Author Response · Author response to Decision Letter 0]

16 Apr 2020

Reviewer #1: The main topic is interesting and could result in the retrievability of implant-retained cemented crowns, which is very interesting considering clinical practice. Although the journal guidelines do not have restrictions on word count, number of figures, or amount of supporting information, it states that authors are encouraged to present and discuss their findings concisely. Thus, my main consideration regarding this manuscript is that it should be thoroughly revised in order to provide only essential information to the readership. The way it is presented is too long and sometimes repetitive; some parts of the introduction should be moved to the discussion section and also some parts of the discussion section should be removed or revised in order to be more concise.

RESPONSE: The authors appreciated this comment and agreed with the reviewer that the manuscript should be more concise and focused. The Introduction and Discussion were shortened. Irrelevant information has been removed.

TEXT CHANGES: The Introduction, which was about 4 pages long, is now shortened in half (now just a little over 2 pages). The Discussion was previously about 7 pages. Now the Discussion is about 5 and 1/2 pages.

Regarding to the presentation of the results, Table 1 does not provide a title and it does not present which outcome was measured and compared in it.

Figures 1A and 1B does not add much to the text. I suggest the authors create a scheme or use another picture to illustrate the use of the laser to remove the crowns. Also, the high number of figures should be revised by the authors. I think that there is no need for all those SEM and EDS images since no significant difference was observed when groups were compared.

RESPONSE: We appreciate the comments and suggestions.

TEXT CHANGES: The table 1’s title was added. We condensed all 9 figures into 5 figures and removed unnecessary SEM figures as suggested. Figure 1A and 1B have been removed. All Figures were revised and condensed. Figure 1 is not a scheme of crown removal and SEM/EDS. Figure 2 is SEM for abutments. Figure 3 is SEM for crowns. Figure 4 is EDS analysis for abutments. Figure 5 is EDS analysis for crowns.

Reviewer #2: Thank you for considering PLOS ONE to publish your work. Authors explored the use of two different types of

pulsed erbium lasers as a non-invasive tool to retrieve cemented zirconia crowns from zirconia implant abutments. At one of the lasers authors explored additionally its use at a second removal of the crown that was recemented. Basically the manuscript deals with an interesting topic, clinically relevant, the methods seems sound, the results interesting, and authors did a great job discussing and introducing the theme. By that, in my humble opinion, the manuscript has potential to be published as it contributes to the knowledge of the theme. Despite that some aspects should be considered further, prior to acceptance:

Major aspects:

- PLOS ONE requires that authors to enable access to the data, authors answered that the data is within the manuscript, however only mean and standard deviation could be found on table1. Please make the data available, or attempt on justify its absence.

RESPONSE: We place the raw data on the removal time as well as full statistical analysis as Supplementary data.

TEXT CHANGES: Information for Supplementary data was added and cited in the manuscript.

- Authors should at least justify why exploring two scenarios of crown retrieval using one laser only, and not with both methods under study.

RESPONSE: We truly appreciate this comment. The reason why we did not perform an equal experiment for both Erbium lasers was that at the experimental design state, we had a lot of experience with the Er:YAG laser in debonding implant crowns. However, we do not have any experience on the Er,Cr:YSGG laser. We were not certain that Er,Cr:YSGG would work in the similar way as Er:YAG. We do plan to do more experiments with different materials and repeated cementations on Er,Cr:YSGG laser in the future.

TEXT CHANGES: The following clarification statement was added into the Discussion.

“Note that the reason why an equal experiment for both Erbium lasers was not planned was there were more previous experimental data and research on the Er:YAG laser in implant crown removal. However, at the time of designing this experiment there was no information in the literature on the Er,Cr:YSGG laser implant crown removal. The investigators were not certain if Er,Cr:YSGG would work in the similar way as Er:YAG. Certainly, there will be in the future, more experiments with different materials and repeated cementations on Er,Cr:YSGG laser implant crown removal.”

- One important aspect is in regards of sample size estimation. It can be seen that using G1 and G2 the Standard deviation was higher than at G3. Perhaps, with a large sample size it would be enable to see statistical differences. It is understandable that clinically a time ranging from 5-6min did not correspond to great/ clinically relevant differences. Despite that, it is necessary to justify the sample size by means of an estimation or a power analysis, and discuss such aspect.

RESPONSE: We appreciate this kind comment. We calculated our sample size from our previous study (Deeb et al 2019 PLoS ONE) using the means of 282 sec and 192 sec (two groups with maximal differences) and the estimated standard deviation of 56.92. With α=0.05 and desired power of 0.80, the sample size would be 6. However, with this type of experiment, sample size calculation may not be appropriate. And we traditionally need at least sample size of 10. Using the same means and standard deviation at α=0.05 and sample size of 10, we will have power of 0.98. So, we should have sufficient statistical power for this sample size. However, we do think that it is important to address the much larger standard deviation values. While the G1 groups had standard deviations of 42.90 and 42.54 and G2 had 26.27, these values are not really representing a true continuous measure. The reason is that we irradiated the sample first for 120 sec then followed with 30 sec for each removal attempts. The time difference is therefore in an increment of 30 sec. The difference between the 42 and 26 values are representing about ~1 attempt difference.

TEXT CHANGES: N/A

- The researcher that attempt on remove the crown was blind for the group? Authors describes as pulling action and gentle tapping, which is rather unclear. There was any standardisation of the load applied during this procedure? Or any assurance of been applied at the same manner for all groups.

RESPONSE: Attempts were made to standardize the laser irradiation as well as crown removal process. However, we did not blind the investigators since we know what each laser looked like. Only one investigator (KGL) who was an expert on Erbium lasers use laser irradiation. And only one investigator (AE) attempted to remove the crown. Another investigator (JGD) measured and noted the time of removal. While we were not blind, each investigator was only doing one procedure in a standardized manner.

TEXT CHANGES: A clarification statement was added in the Methods.

“Attempts were made to standardize the laser irradiation and crown removal process. For each experiment, same three investigators performed each of the following procedures for each crown removal. One investigator did all laser irradiation. One investigator attempted to remove the crown. And one investigator recorded the removal time.”

- Authors expressed that the Y-TZP used is partially 3Y-TZP, and partially 5Y-TZP. 3Y-TZP is known as a partially stabilised zirconia, and by that phase transformation may happen, and potentially low-temperature degradation through time. Thus, one important aspect in regards of potential structural alterations promoted by laser application would be it triggering phase transformation. Even that authors did not access such outcome by raman or X-RD analysis, this aspect should be at least discussed as a limitation. OR the partially composition of 5Y-TZP makes this material fully stable and the phase transformation did not takes place?

RESPONSE: This is an important point. We truly appreciate this comment. 

TEXT CHANGES: The following statement was added in the Discussion in the limitations of the study. 

“In this study, a combination of 3Y-TZP and 5Y-TZP zirconia was used. It is possible that there may be some phase transformation and low temperature degradation. Future studies with Raman spectroscopy or X-Ray Diffraction (XRD) analysis should be done to address this issue.”

Minor aspects:

- At the abstract the groups description is very confusing. At the manuscript material and methods section is much more clear. Please review.

RESPONSE: We appreciate this insight. 

TEXT CHANGES: The Abstract was amended. The Methods in the Abstract is now read:

“Twenty identical zirconia crowns were cemented onto 20 identical zirconia prefabricated abutments using self-adhesive resin cement. The specimens were divided into two groups for laser assisted crown removal; G1 for erbium-doped yttrium aluminum garnet laser (Er:YAG), and G2 for erbium, chromium-doped yttrium, scandium, gallium and garnet (Er,Cr:YSGG). For the G1, after the first crown removal, the specimens were re-cemented and removed again using the Er:YAG laser. Times needed to remove the crowns were recorded and analyzed using ANOVA (α=0.05). The surfaces of the crown and the abutment were further examined using scanning electron microscopy (SEM) and energy-dispersive X-ray spectroscopy (EDS) analyses.”

- Figures, if possible to add group codenames on figures i believe it would facilitate for the reader. Now it is only present on Figure legend description.

RESPONSE: We thank the reviewer for this kind suggestion.

TEXT CHANGES: We added group codenames as well as surface names. We also added arrows in the area of interests.

Figures, the amount of figures seems to be excessive. All of these figures are really necessary?

RESPONSE: We thank the reviewer for this kind suggestion.

TEXT CHANGES: We condensed the figures from 9 to 5 and removed unnecessary figures from the manuscript.

---

## [Decision Letter · Decision Letter 1]

8 May 2020

Exploring the use of pulsed erbium lasers to retrieve a zirconia crown from a zirconia implant abutment

PONE-D-20-08097R1

Dear Dr. Bencharit,

We are pleased to inform you that your manuscript has been judged scientifically suitable for publication and will be formally accepted for publication once it complies with all outstanding technical requirements.

With kind regards,

Rafael Sarkis-Onofre

Academic Editor

PLOS ONE

Additional Editor Comments (optional):

Reviewers' comments:

Reviewer's Responses to Questions

**Comments to the Author**

1. If the authors have adequately addressed your comments raised in a previous round of review and you feel that this manuscript is now acceptable for publication, you may indicate that here to bypass the “Comments to the Author” section, enter your conflict of interest statement in the “Confidential to Editor” section, and submit your "Accept" recommendation.

Reviewer #1: All comments have been addressed

Reviewer #2: All comments have been addressed

2. Is the manuscript technically sound, and do the data support the conclusions?

Reviewer #1: Yes

Reviewer #2: Yes

3. Has the statistical analysis been performed appropriately and rigorously? 

Reviewer #1: Yes

Reviewer #2: Yes

4. Have the authors made all data underlying the findings in their manuscript fully available?

Reviewer #1: Yes

Reviewer #2: Yes

5. Is the manuscript presented in an intelligible fashion and written in standard English?

Reviewer #1: Yes

Reviewer #2: Yes

6. Review Comments to the Author

Reviewer #1: (No Response)

Reviewer #2: Thank you for consider and successfully address all prior raised criticism. Based on that i recommend the acceptance.

7. PLOS authors have the option to publish the peer review history of their article (what does this mean?). If published, this will include your full peer review and any attached files.

Reviewer #1: Yes: Mateus BF dos Santos

Reviewer #2: No

---

## [Editor Report · Acceptance letter]

22 May 2020

PONE-D-20-08097R1 

Exploring the use of pulsed erbium lasers to retrieve a zirconia crown from a zirconia implant abutment. 

Dear Dr. Bencharit:

I am pleased to inform you that your manuscript has been deemed suitable for publication in PLOS ONE. Congratulations! Your manuscript is now with our production department. 

With kind regards,

on behalf of

Dr. Rafael Sarkis-Onofre 

Academic Editor

PLOS ONE